# Comparison of the Virulence of Space Mutants of *Aspergillus oryzae* XJ-1 against Adult *Locusta migratoria*

Xin Fu [1,2], Hui Liu [1,3], Xiao Xu [1,3], Jin Guo [1], Shaojing Hu [1], Yinwei You [1,3,*] and Long Zhang [1,3,*]

1   Institute of Plant Protection, Shandong Academy of Agricultural Sciences, Jinan 250100, China; fuxin0056@gmail.com (X.F.); liuhui6892@gmail.com (H.L.); xuxiao_cau@yeah.net (X.X.); guojin1706@gmail.com (J.G.); hushaojing12@gmail.com (S.H.)
2   College of Grassland Science and Technology, China Agricultural University, Beijing 100083, China
3   Shandong Provincial Engineering Technology Research Center on Bio-Control of Crops Diseases and Insect Pest, Jinan 250100, China
*   Correspondence: youyinwei88@gmail.com (Y.Y.); locust@cau.edu.cn (L.Z.); Tel.: +86-531-66655309 (Y.Y. & L.Z.)

**Abstract:** Biological control methods provide a sustainable approach for reducing agricultural losses caused by locust plagues. Space mutagenesis can generate high numbers of mutations using satellites and spacecrafts, including beneficial and stable mutants. *Aspergillus oryzae* XJ-1 was recently reported to show high virulence against locusts. We subjected this fungal pathogen to space mutagenesis to obtain more effective strains. Pathogen conidia powder was mutated in the China Space Station for 6 months. We obtained five mutants of *A. oryzae* XJ-1, TQ201, TQ238, TQ302, TQ549, and TQ555, and all mutants were identified as *A. oryzae* by molecular techniques. TQ549 showed the highest virulence against adult *L. migratoria* ($LT_{50}$: $4.97 \pm 0.21$ days); the $LT_{50}$ of *A. oryzae* XJ-1 was $5.67 \pm 0.06$ days. Both TQ549 and *A. oryzae* XJ-1 grew most rapidly at 33 °C on potato dextrose agar (PDA) plates. There was no significant difference in the growth rate of TQ549 and *A. oryzae* XJ-1 at 24 °C. The colony morphological characteristics of the five mutants on PDA plates differed from that of *A. oryzae* XJ-1. The space mutant TQ549, which showed high virulence against adult locusts, could be used as a biological control agent for the control of locust infestations.

**Keywords:** space mutagenesis; *Aspergillus oryzae*; locust; virulence; growth; mortality; temperature

## 1. Introduction

Locusts are economically significant agricultural pests, and the livelihoods of approximately 1/10 of the human population are affected by locusts. Locust outbreaks are responsible for major losses of crops on an annual basis [1–3]. Chemical pesticides have been the main method for the control of locusts since the 1940s, but pesticide residues can have negative effects on the health of humans and the environment [1,4].

Environmentally friendly biopesticides have been widely applied [5]. Some microorganisms, such as protozoa (e.g., *Nosema locustae*), fungi (e.g., *Metarhizium* spp. and *Beauveria* spp.), and viruses (e.g., entomopoxvirus), have been reported to be important pathogens of locusts and grasshoppers [3,6–15], and protozoa and fungi have been widely used as microbial control agents; they have been shown to be effective alternatives to chemical pesticides for the control of locust infestations [1,14]. These two microbial control agents could induce the mortality of 60–80% of locust nymphs, but their effectiveness against adult locusts is poor. It is necessary to develop a new microbial control agent that has higher virulence to adult locusts.

Generally, *Aspergillus oryzae* has been one major component of the leaven in traditional fermented food such as soy sauce, soybean paste and rice wine in China, Japan, and other Asian countries for centuries [16]. *A. oryzae* does not produce the most potent natural carcinogen, aflatoxin, or any other carcinogenic metabolites. It is listed as Generally

Recognized as Safe (GRAS) by the Food and Drug Administration (FDA) in the USA. The pathogenicity of *Aspergillus* to locust is rarely reported. *A. flavus* was reported to have high virulence to the desert locust *Schistocerca gregaria* in 1975, but until now, no further research was reported [17]. We had isolated and identified a new fungal locust pathogen *A. oryzae* XJ-1 from a dead locust collected in the Chonghuer grassland of Altay Prefecture, Xinjiang Uyghur Autonomous Region, China, in 2011, which was highly virulent in third-instar nymphs of *L. migratoria* in a laboratory study [18]. Recently, we found that *A. oryzae* XJ-1 could also effectively control adult locust infestations in both laboratory experiments and field trials [19].

The main factor limiting the efficacy of most microbial control agents is that locusts are not rapidly killed, and mortality is often not achieved until 10 days after application, or even longer [1,2,4,6–11]. Locusts can grow and develop rapidly and form heavy swarms [1,3]. Adult locust swarms can migrate long distances; for instance, desert locust adult swarms can migrate more than 150 km/day [2]. Achieving rapid control of adult locust populations remains a major challenge in the biological control of locust plagues.

Physical and chemical mutagenesis [20] and space mutagenesis [21] have been used to generate mutant microorganisms for screening highly efficient strains. Space mutagenesis can generate high numbers of mutations and yield variable and stable mutants [22]. The vast, cold, and radiation-filled conditions of outer space pose major challenges for all organisms [23]. Microorganisms rapidly adapt to environmental changes by altering the expression of their genes [24]. A variety of mutants of bacteria, fungi, animals, and plants have been generated by space mutagenesis using satellites and spacecrafts. To obtain strains with greater virulence, we conducted space mutagenesis of the conidia powder of *A. oryzae* XJ-1 in the China Space Station; we then screened the mutants based on the morphological characteristics of colonies on potato dextrose agar (PDA) plates and determined their virulence against adult locusts. We also evaluated the growth rate of the strains at different temperatures on PDA plates in the lab. We obtained a highly virulent space mutant strain, TQ549, which has white colonies and has the potential to be used for the control of locust infestations.

## 2. Materials and Methods

### 2.1. Screening and Identification of A. oryzae XJ-1 Mutants

Space mutagenesis of 5 g of conidia powder of *A. oryzae* XJ-1 was performed on the Shenzhou-14 crewed spaceship from 5 June 2022 to 4 December 2022. A small quantity of conidia powder was suspended in 120 mL of sterile distilled water with 0.3% (*v/v*) Tween-80. A hemocytometer was used to determine the concentration; it was then adjusted to 1000 conidia mL$^{-1}$. Next, 100 μL of conidia suspension was spread on a PDA plate and incubated at 30 °C for 3 days in an incubator. No more than 10 colonies will grow on every PDA plate. A total of 1200 PDA plates were inoculated. Observations of these agar plates were made every day, and mutants were screened by comparing colony shape and color to that of *A. oryzae* XJ-1. These mutants were placed in a freezer at −40 °C for subsequent experiments. Mutants of *A. oryzae* XJ-1 were grown in PDA liquid medium broth for 3 days. The mycelia were harvested and then ground using the liquid nitrogen method. DNA was extracted using the TIANgel Midi Purification Kit (Tiangen, Beijing, China), according to the manufacturer's instructions. The ITS sequences of mutants were amplified using the universal primers ITS1 (TCCGTAGGTGAACCTGCGG) and ITS4 (TCCTCCGCTTATTGATATGC) and the Go Taq® Green Master Mix (Promega Corporation, Madison, WI, USA). The thermal cycling conditions were as follows: initial denaturation at 95 °C for 3 min; 35 cycles of 95 °C for 30 s, 55 °C for 30 s, and 72 °C for 60 s; and a final extension of 72 °C for 10 min. These ITS sequences were cloned in the pGEM-T vector using the pGEM®-T Vector System I (Promega Corporation, Madison, WI, USA) and sequenced. This process was repeated 3 times for each strain. Multiple sequence alignment was performed using BioEdit 7.0.9 software to analyze the relationships between the mutants and *A. oryzae* XJ-1 (GenBank: KP067321).

*2.2. Comparison of the Virulence of A. oryzae Mutants and XJ-1 against Adult L. migratoria*

*L. migratoria* adults were reared in the Biocontrol Laboratory of the Institute of Plant Protection at the Shandong Academy of Agricultural Sciences. Adult locusts were tested 2–8 days after eclosion. Locusts were fed fresh wheat seedlings daily at 28–30 °C and 60% relative humidity and under an 18 h/6 h light/dark photoperiod. Feces were regularly removed from the locust enclosures.

A bioassay experiment was preliminarily conducted to screen mutants with high virulence, mainly according to You et al. (2023) [19]. Briefly, a total of 100 healthy adults of *L. migratoria* were equally divided into five groups for virulence bioassays of TQ201, TQ238, TQ302, TQ549, and TQ555. The tested locusts were inoculated by individually immersing them in $10^6$ conidia $mL^{-1}$ suspension in Petri dishes for less than 1 s. All treated locusts were dried and then individually reared after treatment in a plastic box with a round top (diameter, 14 cm), round bottom (diameter, 9 cm), and height of 14. Dead locusts were collected and recorded daily for 15 days after treatment. Another bioassay was performed to compare the virulence of TQ549 and *A. oryzae* XJ-1. A total of 60 healthy adults of *L. migratoria* were equally divided into three groups and inoculated with $10^6$ conidia $mL^{-1}$ suspension of TQ549, *A. oryzae* XJ-1, and 0.3% (vol/vol) Tween-80 solution, which was used as a control. Three replications of all experiments were performed. The virulence of *A. oryzae* XJ-1 and mutants was analyzed. An infection experiment was performed as described in You et al. (2023) [19]. Briefly, dead locusts inoculated with *A. oryzae* XJ-1 and mutants were collected in the bioassay experiment. Each dead locust was placed on a sterilized glass slide on a sterilized filter paper in a sterilized Petri dish, and 200 μL of sterile water was added to soak the filter paper. The Petri dish was placed in an incubator at 28 °C for 7 days. Every two days, the growth of *A. oryzae* XJ-1 and the mutants on the locusts was observed and recorded. We used these two methods to demonstrate the virulence of the mutants.

*2.3. Comparison of the Optimal Growth Temperature of TQ549 and A. oryzae XJ-1*

One μL of conidia suspension of TQ549 and *A. oryzae* XJ-1 was inoculated on the center of a PDA plate and incubated at 24, 26, 28, 30, 32, 33, 34, 35, 36, and 38 °C in incubators. The growth rates of TQ549 and *A. oryzae* XJ-1 were evaluated by measuring colony diameters every day with a ruler from the back of culture dishes until the 8th day after inoculation. Three replications of the experiments were performed.

*2.4. Statistical Analysis*

All analyses were conducted in Origin 2022b software (OriginLab, Northampton, MA, USA). One-way ANOVA followed by Tukey's post hoc test was used to evaluate differences in the corrected mortality and the median lethal time ($LT_{50}$) between TQ549 and *A. oryzae* XJ-1. A Student's *t*-test was used to analyze differences in the colony diameter between groups. $LT_{50}$ was determined using probit (logit) analysis with Excel 2016 Software (Microsoft, Redmond, WA, USA).

**3. Results**

*3.1. Screening and Identification of XJ-1 Mutants*

Five mutants were obtained from 564 colonies developed from a monospore following treatment with space mutagenesis. The colonies of TQ201, TQ238, and TQ302 were fluffier than the colony of *A. oryzae* XJ-1 on the PDA plate, and the conidial color of these three mutants was all green, just like *A. oryzae* XJ-1; the conidial color of TQ549 and TQ555 was white and golden yellow on the PDA plate, respectively, whereas the conidial color of *A. oryzae* XJ-1 was green (Figure 1). These five mutants could infect and grow in adult locusts; they eventually covered the bodies of dead locusts, especially TQ549, and as a result they mainly exhibited a white color (Figure 1). To further identify whether these mutants were *A. oryzae* XJ-1 mutants, the ITS sequences of these five mutants were amplified by PCR, then cloned and sequenced (Figure 2). The ITS sequences of *A. oryzae* XJ-1 and its mutants

were all 595 bp. These ITS sequences were deposited in the GenBank database (accession numbers: OR866213.1, OR878662.1, OR875384.1, OR878663.1, and OR878664.1). The ITS sequences of the five mutants were compared with that of *A. oryzae* XJ-1 (GenBank accession number: KP067321). The ITS sequence of TQ302 differed from that of *A. oryzae* XJ-1 by three base pairs, and the sequence similarity was 99.50%; however, only two base pair differences were detected between the ITS sequences of the other four mutants and that of *A. oryzae* XJ-1, and their sequence similarities were all 99.66% (Table 1, Supplementary Figure S1). These findings indicated that these five mutants were all *A. oryzae* XJ-1 mutants generated by space mutagenesis.

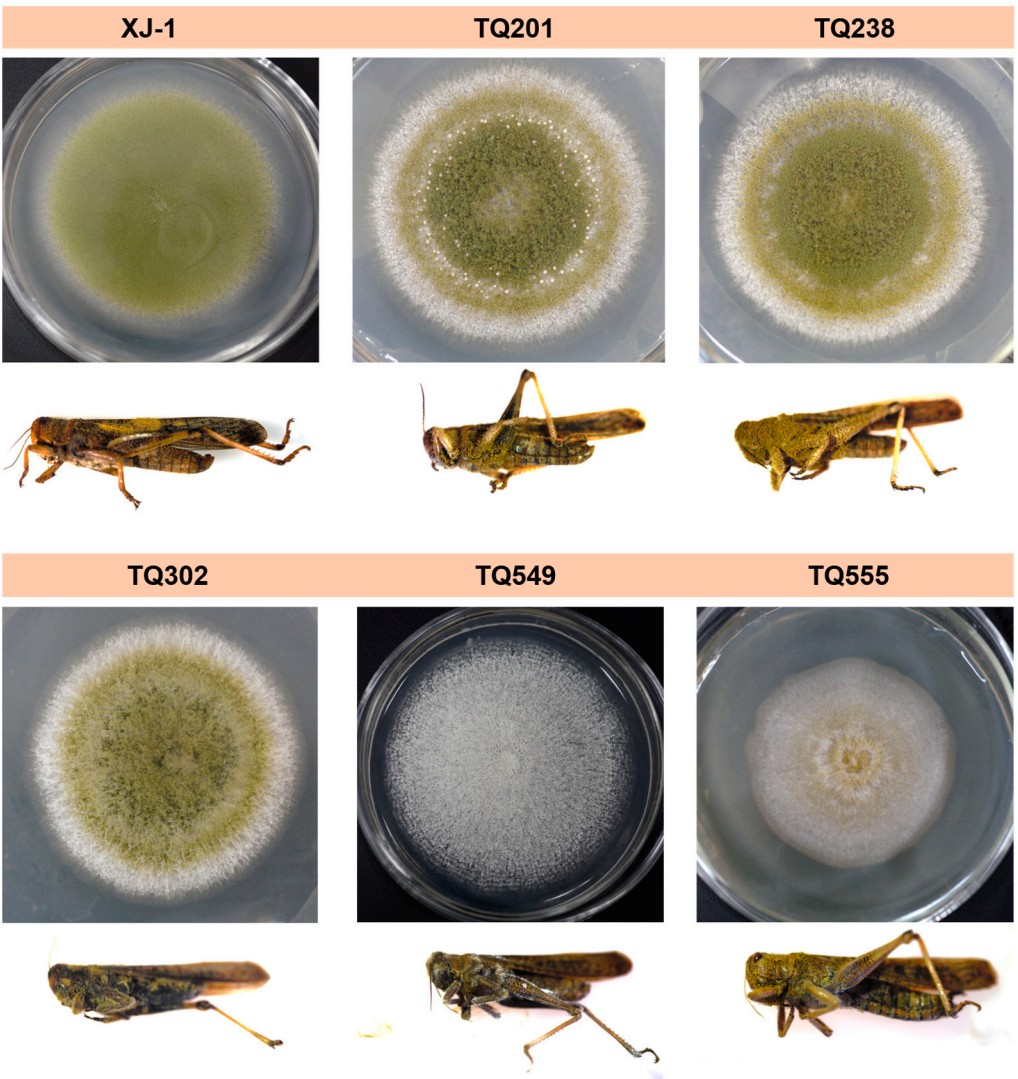

**Figure 1.** The morphological characteristics of the colonies of mutants and *A. oryzae* XJ-1 in PDA plates and infected locust adults.

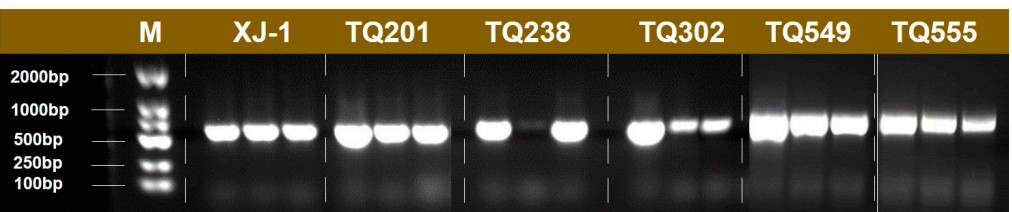

**Figure 2.** The colony PCR of ITS sequences of five mutants and *A. oryzae* XJ-1. Three colony PCR reactions were performed for each sample.

**Table 1.** Molecular identification of five mutants by ITS sequences.

| Strain | Length (bp) | Result |
| --- | --- | --- |
| TQ201 | 595 | 99.66% identity with *A. oryzae* XJ-1 |
| TQ238 | 595 | 99.66% identity with *A. oryzae* XJ-1 |
| TQ302 | 595 | 99.50% identity with *A. oryzae* XJ-1 |
| TQ549 | 595 | 99.66% identity with *A. oryzae* XJ-1 |
| TQ555 | 595 | 99.66% identity with *A. oryzae* XJ-1 |

### 3.2. Virulence of Mutants against L. migratoria Adults in the Laboratory

We examined the virulence of five mutants against *L. migratoria* adults by inoculation at a concentration of $10^6$ conidia mL$^{-1}$. TQ549 induced the highest cumulative mortality within the shortest time as the curve of mortality peaked at $96.67 \pm 1.67\%$ on the 9th day after inoculation. The cumulative mortality of locusts caused by TQ201 was second to that caused by TQ549 and peaked on the 15th day after inoculation. The mortalities of locusts caused by the other three mutants were not over 95%, even on the 15th day in the same experimental period. The cumulative mortality of the adult locusts caused by TQ302 was the lowest and peaked at $50.00 \pm 5.78\%$ on the 15th day after inoculation (Figure 3). We also conducted an experiment to compare the virulence of TQ549 and *A. oryzae* XJ-1. The corrected cumulative mortalities of TQ549 on the 6th ($72.71 \pm 3.24\%$), 7th ($81.01 \pm 2.33\%$), 8th ($86.49 \pm 2.08\%$), and 9th ($90.41 \pm 1.86\%$) days after treatment were significantly higher than those of wild type, *A. oryzae* XJ-1, on the 6th ($56.34 \pm 0.78\%$), 7th ($67.70 \pm 2.99\%$), 8th ($74.95 \pm 2.21\%$), and 9th ($82.68 \pm 0.33\%$) days, respectively (Figure 4A). The LT$_{50}$ of TQ549 ($4.97 \pm 0.21$ days) was significantly lower than that of *A. oryzae* XJ-1 ($5.67 \pm 0.06$ days), indicating that TQ549 caused the death of locust adults more rapidly than *A. oryzae* XJ-1, and the virulence of TQ549 was higher than that of *A. oryzae* XJ-1 (Figure 4B).

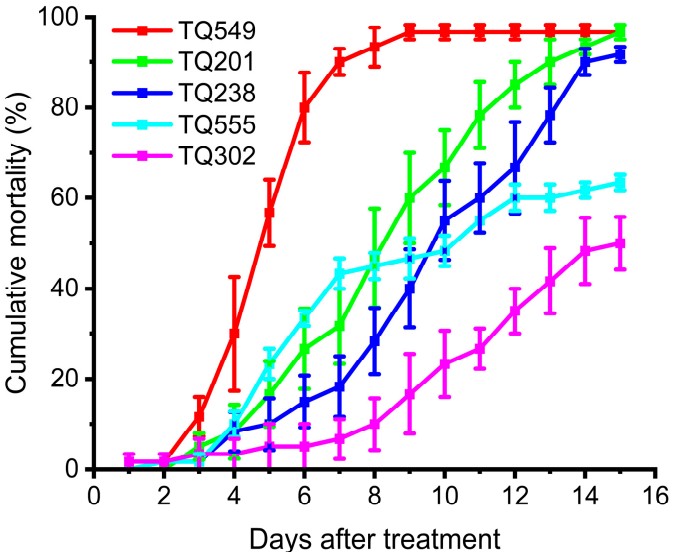

**Figure 3.** The cumulative mortalities of locust adults caused by inoculation with mutants at a concentration of $10^6$ conidia mL$^{-1}$. Bar: SE.

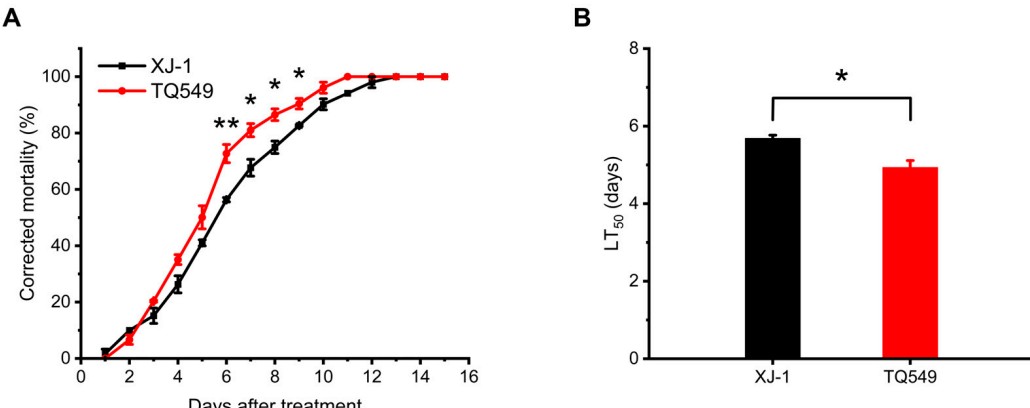

**Figure 4.** Comparison of the virulence of TQ549 and *A. oryzae* XJ-1 against locust adults. (**A**) Corrected mortalities of two strains against locust adults at a concentration of $10^6$ conidia $mL^{-1}$. Asterisk indicates significant difference between treatments on the same day, according to One-way ANOVA comparison with Tukey's multiple comparisons test. *, $p < 0.05$, **, $p < 0.01$. Bar: SE; (**B**) $LT_{50}$ of two strains. Bar: SE. One-way ANOVA comparison with Tukey's multiple comparisons test. $p = 0.019$, *, $p < 0.05$, **, $p < 0.01$.

### 3.3. Comparison of the Growth Rates of TQ549 and A. oryzae XJ-1 at Different Temperatures

The diameters of the colonies of both TQ549 and *A. oryzae* XJ-1 peaked at 33 °C within 5–6 days among the various temperatures tested (24, 26, 28, 30, 32, 33, 34, 35, 36, and 38 °C); however, at other temperatures, more time was required for the peak diameter to be achieved, indicating that 33 °C was the optimal growth temperature for the two strains on the PDA plate (Figure 5). No significant differences in growth between TQ549 and *A. oryzae* XJ-1 were observed on all days at 24 °C (Table 2). However, significant differences in their growth were observed on the 3rd, 4th, and 5th days after inoculation at 33 °C, indicating that TQ549 grew slower than *A. oryzae* XJ-1 during the middle of the experimental period at 33 °C; however, no significant differences in growth were observed after this period (Table 2). Significant differences in the growth of the two strains were observed on the 2nd, 3rd, 4th, and 8th days after inoculation at 38 °C (Table 2).

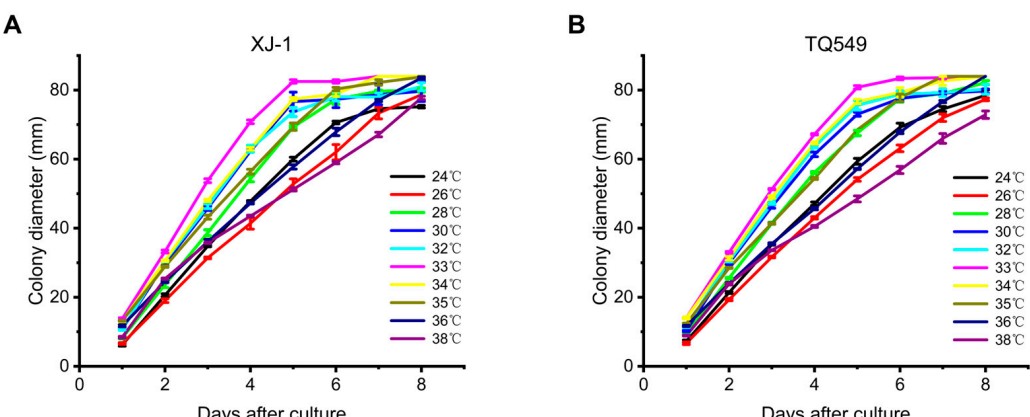

**Figure 5.** Growth rates of *A. oryzae* XJ-1 (**A**) and TQ549 (**B**) at different temperatures, as indicated by the colony diameter. Bar: SE.

**Table 2.** Comparison of the colony diameters of TQ549 and *A. oryzae* XJ-1 after inoculation from the 1st to 8th day at 24 °C, 33 °C, and 38 °C.

| Days | 24 °C | | 33 °C | | 38 °C | |
|------|-------|--|-------|--|-------|--|
| | *A. oryzae* XJ-1 | TQ549 | *A. oryzae* XJ-1 | TQ549 | *A. oryzae* XJ-1 | TQ549 |
| 1 | 6.20 ± 0.37 a | 7.25 ± 0.37 a | 13.85 ± 0.31 a | 14.00 ± 0.22 a | 8.44 ± 0.26 a | 8.88 ± 0.13 a |
| 2 | 20.60 ± 0.40 a | 21.38 ± 0.26 a | 33.25 ± 0.41 a | 33.00 ± 0.26 a | 25.38 ± 0.18 a | 23.88 ± 0.31 b |
| 3 | 34.80 ± 0.20 a | 35.25 ± 0.16 a | 53.75 ± 0.56 a | 51.30 ± 0.20 b | 35.75 ± 0.16 a | 33.63 ± 0.13 b |
| 4 | 47.80 ± 0.20 a | 47.13 ± 0.48 a | 70.81 ± 0.53 a | 67.20 ± 0.20 b | 43.50 ± 0.27 a | 40.50 ± 0.29 b |
| 5 | 60.00 ± 0.55 a | 59.38 ± 0.84 a | 82.50 ± 0.50 a | 80.80 ± 0.49 b | 51.25 ± 0.46 a | 48.50 ± 0.87 a |
| 6 | 70.60 ± 0.40 a | 69.25 ± 1.17 a | 82.50 ± 0.50 a | 83.40 ± 0.40 a | 59.00 ± 0.34 a | 56.87 ± 1.05 a |
| 7 | 74.60 ± 0.51 a | 74.50 ± 0.73 a | 84.00 a | 83.60 ± 0.10 a | 67.13 ± 0.67 a | 66.00 ± 1.41 a |
| 8 | 75.20 ± 0.37 a | 78.50 ± 1.51 a | 84.00 a | 84.00 a | 77.75 ± 0.49 a | 72.88 ± 1.09 b |

Note: Different characters after the numbers indicate significant differences between TQ549 and *A. oryzae* XJ-1 on the same day and at the same temperature, according to *t*-tests.

## 4. Discussion

Five mutants were screened from 564 strains of *A. oryzae* XJ-1 generated by space mutagenesis. The colony morphological characteristics of these five mutants on PDA plates differed from that of *A. oryzae* XJ-1. Identification of these mutants according to analysis of their ITS sequences revealed that they belonged to *A. oryzae*. Experiments aimed at evaluating the mortality rates of these mutants and *A. oryzae* XJ-1 against *L. migratoria* adults showed that only one white colony of the mutant TQ549 exhibited higher virulence than the original strain *A. oryzae* XJ-1; the virulences of the other four mutants were lower than that of *A. oryzae* XJ-1. Changes in virulence may stem from the radiation-filled conditions of outer space [23]. In the filamentous fungus *Aspergillus fumigatus*, low-nutrient environments combined with enhanced irradiation and microgravity might trigger changes in the molecular suite of microorganisms with increased virulence and resistance to microbes [25]. Pathogenicity genes of the mutants might contribute to the observed changes in the mutants. Five mutants had different virulence against adult locusts, indicating that different pathogen-related genes might be mutated in every mutant. So, these mutants were good materials for exploring the pathogenic molecular mechanism of *A. oryzae* XJ-1. Comparison of the genes in these mutants and *A. oryzae* XJ-1 coupled with molecular biological methods could provide insights into the molecular mechanism by which *A. oryzae* XJ-1 infects locusts. Such studies could shed light on how genes could be modified to increase the virulence of *Aspergillus* spp., which could aid the development of more effective biological control agents for the control of locusts.

In our experiment, two colonies of mutants on PDA plates were acquired with different conidial colors, TQ549 (white) and TQ555 (golden yellow), and changes in their pathogenicity were detected. The virulence of TQ549 was higher than that of *A. oryzae* XJ-1; however, TQ555 was lower. This is consistent with the results of previous studies showing that conidial color might affect the pathogenicity of *A. flavus* [26] and *A. fumigatus* [27–29]. However, additional studies are needed to clarify the relationships between the color and pathogenicity of *Aspergillus* spp., including for our mutants. When we cultured the mutants and *A. oryzae* XJ-1 on PDA plates, we found TQ555 grew much slower than the other four mutants and *A. oryzae* XJ-1. Low growth rate of TQ555 may be one reason for low virulence against adult locusts.

Adaptation to a wide range of temperatures is important for the control of locusts, because locusts are widely distributed in tropical and temperate regions. Our experiment on the effect of temperature on *A. oryzae* XJ-1 and the space mutant TQ549 showed that the optimal growth temperature was 33 °C for both TQ549 and *A. oryzae* XJ-1. TQ549 and *A. oryzae* XJ-1 could grow relatively rapidly on a PDA plate at 24 and 38 °C, indicating that these two strains might be adapted to a wide temperature range. The temperatures of central and southern China when locusts are most active often exceed 35 °C. *A. oryzae* XJ-1 had been applied for the control of locusts at temperatures ranging from 26 to 38 °C in

China during 2022 in a large-scale field experiment, and the control efficacy of *A. oryzae* XJ-1 under these conditions was high [19]. Therefore, TQ549 might be effective for the control of locusts over a wide temperature range as well.

The space mutant TQ549 caused the highest cumulative mortality within the shortest time when *L. migratoria* adults were inoculated at a concentration of $10^6$ conidia $mL^{-1}$ in five mutants; the mortality was over 80% on the 6th day and reached the peak at $96.67 \pm 1.67\%$ on the 9th day after inoculation. By comparing the corrected cumulative mortality and the $LT_{50}$ between TQ549 and wild type, *A. oryzae* XJ-1, we can conclude that the mutant TQ549 could kill locust adults more quickly than *A. oryzae* XJ-1. This is important to overcome the main challenge: that most locust microbial control agents cannot kill locusts rapidly, particularly adult locusts. *Metarhizium* spp. and *P. locustae* are the main biological pesticides used in locust control nowadays, but they need over 10 days to achieve good control efficacy [1,2,4,6–11]. The most notorious migratory desert locust can form heavy adult swarms rapidly and then migrate long distances [1–3]. East Africa suffered from the worst desert locust outbreak in decades in 2019–2020, about 360 billion desert locusts in Ethiopia, Kenya, Somalia, South Sudan, Uganda, and the United Republic of Tanzania [30]. Controlling locust as quickly as possible will not only prevent locust adult swarms from dispersing and infesting large area but also reduce the loss of crops. This mutant provides a possibility to develop a relatively quicker biological control agent as an alternative to chemical pesticide in adult locust control.

Though we have screened a more virulent space mutant, TQ549, as a potential biological control agent against locust adults in laboratory and demonstrated its original fungus, *A. oryzae* XJ-1, effectively controlling locusts in a field trial [18,19], more experiments in the field are needed to develop application methods, such as application rates and conditions, and to understand factors that influence the efficacy of TQ549 in controlling locust adults. Furthermore, we only demonstrated the efficacy of TQ549 on *L. migratoria* adult in this experiment. As we know, the locust has quite a long nymphal developmental stage, including 1–5 or 6 instar nymphs; as such, to save crops, it is also very important to control nymphs. Therefore, more experiments must be conducted to check the efficacy of TQ549 against locust nymphs.

In the world, there are more than 500 species of acridids (Orthoptera: Acridoidea) that can cause damage to pastures and crops, and about 50 are considered major pests [1]. Previous reports indicated that microbial control agents including *N. locustae* and *Metarhizium* could cause different pathogenicity in different species of locusts and grasshoppers [31]. It is necessary to determine the pathogenicity of the mutant TQ549 against each economically important species of grasshopper and locust in the world and particularly those in China, such as *Calliptamus italicus*, *Oedaleus asiaticus*, *Ceracris kiangsu*, *Calliptamus abbreviates*, *Oxya chinensis*, and *Epacromius coerulipes*. This will be very useful for expanding the application of the mutant TQ549 in the future.

In addition, to develop a more sophisticated biological control system in locust and grasshopper whole life cycles, more research is required to determine how the mutant TQ549 may be integrated with *A. oryzae* XJ-1, or *N. locustae*, or *Metarhizium* spp. for better applications according to the features of these previous biological control agents. Previous studies have demonstrated that concurrent use of *N. locustae* and *Metarhizium* spp. could result in additive effects on locusts and grasshoppers [31]. So, this work will help to develop a better integrated locust management system.

## 5. Conclusions

1.  Five mutants were screened from *A. oryzae* XJ-1 generated by space mutagenesis, and the morphological characteristics of their colonies on PDA plates differed from that of *A. oryzae* XJ-1. Analysis of their ITS sequences indicated that all these mutants belonged to *A. oryzae*.
2.  Experiments on the mortality rates of these mutants and *A. oryzae* XJ-1 against *L. migratoria* adults showed that only one mutant, TQ549, exhibited higher virulence

than the original strain, *A. oryzae* XJ-1, and the virulences of the other four mutants were all lower than that of *A. oryzae* XJ-1.

3. Our results indicated that space mutant TQ549, which had white conidia, high virulence against adult locusts, and tolerance of a wide temperature range, had the potential to be used as a biological agent for the control of locusts.

4. These mutants with different virulence against adult locusts will provide us with good materials to research the pathogenic molecular mechanism of *A. oryzae* XJ-1 by comparison of genomes and transcriptomes of *A. oryzae* XJ-1 and the mutants.

**Supplementary Materials:** The following supporting information can be downloaded at https://www.mdpi.com/article/10.3390/agronomy14010116/s1, Figure S1: ITS multiple sequence alignment between five mutants and *A. oryzae* XJ-1.

**Author Contributions:** L.Z. and Y.Y. conceived and designed research; X.F., H.L., X.X., J.G., S.H., Y.Y. and L.Z. conducted the experiments; Y.Y. and L.Z. wrote the manuscript. All authors have read and agreed to the published version of the manuscript.

**Funding:** This work was supported by the National Key R&D Program of China 2022YFD1400500 and the Agricultural Scientific and Technological Innovation Project of Shandong Academy of Agricultural Sciences CXGC2023F04.

**Data Availability Statement:** The data presented in this study are available on request from the corresponding author.

**Acknowledgments:** The authors are thankful for the help of Shuhan Liu, Pengfei Liu, Weijia Song, and Ruipeng Li for screening and bioassay experiments.

**Conflicts of Interest:** The authors declare that they have no conflicts of interest.

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
