# Peer review of "Comparison of the Virulence of Space Mutants of Aspergillus oryzae XJ-1 against Adult Locusta migratoria"

_agronomy, doi:10.3390/agronomy14010116_

Round 1

Reviewer 1 Report

Comments and Suggestions for Authors

 Comments and suggestions are included in the annex

Author Response

Reviewer 1

  1. Line 78. How many repetitions were used?

Answer: Thanks for your question. We inoculated a total of 300 PDA plates. The details have been added in the article in line 78 in clear version.

  1. Line 79. Were mutant colonies monosporic colonies?

Answer: Thanks for your question. Yes, all mutant colonies were monosporic colonies.

  1. Line 81. What characteristics or criteria were used to differentiate mutant colonies? The shape of the colony, the color of the colony, the growth rate of the colony?

Answer: Thanks for your question. We used colony morphological characteristics to screen mutants including the shape, the color, the growth rate of colony, etc.

  1. Line 92. How many repeats were made by each strain of mutant?

Answer: Thanks for your question. Repeat 3 times for each strain. And the details have been added in line 91 and in line 150-151 in clear version.

  1. Line 96. Which female/male ratio was used?

Answer: Previously, our lab had demonstrated that there was no significant difference in the virulence of Aspergillus oryzae XJ-1 against female and male locusts, so we did not distinguish the ratio of female and male locusts here.

  1. Line 103. How many male or female insects, or females and males were used in each group of 20 insects?

Answer: Thank you very much. Same reason for question 5. We did not distinguish the ratio of female and male locusts here.

  1. Line 106. Which drying technique was used?

Answer: Thank you very much. We use natural air drying to dry the insects.

  1. Line 107. Which food was supplied during the incubation or rearing period?

Answer: Thank you very much. We rear locusts with fresh wheat every day.

  1. Line 110. Which scale, unit of measurement or technique was used to measure virulence?

Answer: Thank you very much. We calculated the corrected mortality of locusts caused by TQ549 and Aspergillus oryzae XJ-1 to measure the virulence. And we calculated the LT50 to compare the virulence between TQ549 and Aspergillus oryzae XJ-1.

  1. Line 119. Which sex were the insects?

Answer: Thank you very much. Same reason for question 5. We did not distinguish the ratio of female and male locusts here.

Reviewer 2 Report

Comments and Suggestions for Authors

Fu et al. investigated the mutants of Aspergillus oryzae XJ-1 against adults of Locusta migratoria. This study proves that one of the mutants presented a higher mortality effect than the XJ-1. Despite the lack of elucidation of which mutagenesis the authors founded in this study, it offers novelty information worth publication.

I have only a minor suggestion on the presentation of results, which needs to include the statistical results of the data. For instance, the median lethal time is reported without the test results (e.g., Chi-square, p-value, and degree of freedom). Also, which estimation method was used in the test is not reported (e.g., Kaplan-Meier), nor how the curves were compared (e.g., log-rank). The LT50 is a non-parametric estimation, using test t, which seems inadequate for that report. 

Author Response

Reviewer 2

Fu et al. investigated the mutants of Aspergillus oryzae XJ-1 against adults of Locusta migratoria. This study proves that one of the mutants presented a higher mortality effect than the XJ-1. Despite the lack of elucidation of which mutagenesis the authors founded in this study, it offers novelty information worth publication.

Answer: Thank you very much.

I have only a minor suggestion on the presentation of results, which needs to include the statistical results of the data. For instance, the median lethal time is reported without the test results (e.g., Chi-square, p-value, and degree of freedom). Also, which estimation method was used in the test is not reported (e.g., Kaplan-Meier), nor how the curves were compared (e.g., log-rank). The LT50 is a non-parametric estimation, using test t, which seems inadequate for that report.

Answer: I appreciate your suggestion very much. Based on your suggestions, we have added the relevant details of the data statistical results. We modified the test method of LT50 (Figure 2B) to One-way ANOVA and chose Tukey's multiple comparisons test. The P-value is 0.019. The method of estimation used in the test was the cumulative mortality curve, and the comparison method was also One-way ANOVA comparison with Tukey's multiple comparisons test.

We changed the sentence into “One-way ANOVA followed by Tukey’s post hoc test was used to evaluate differences in the corrected mortality and the median lethal time (LT50) between TQ549 and A. oryzae XJ-1. A Student's t-test was used to analyze differences in the colony diameter between groups” in line 121-124 in clear version.

We changed the sentence into “Figure 4. Comparison of the virulence of TQ549 and A. oryzae XJ-1 against locust adults. (A) Corrected mortalities of two strains against locust adults at a concentration of 106 conidia ml-1. Asterisk indicates significant difference between treatments on the same day according to One-way ANOVA comparison with Tukey's multiple comparisons test. *, P<0.05, **, P<0.01. Bar: SE; (B) LT50 of two strains. Bar: SE. One-way ANOVA comparison with Tukey's multiple comparisons test. P=0.019, *, P<0.05, **, P<0.01." in line 172-177 in clear version.

Reviewer 3 Report

Comments and Suggestions for Authors

The development of new insect management methodologies is an important part of the continued production of high-quality agricultural products. Developing a new biocontrol organism through enhanced mutation is certainly worth exploring.

There are a few things that should be covered in the introduction to make this paper easier and more informative to readers.

Q1) Is space radiation less harmful than other radiation sources? Is there something about Space that makes this approach worth the extra cost versus exposing them to a radiation source closer to home?

Q2) This is the first I have seen Aspergillus oryzae introduced as an insect biocontrol agent. Sometimes I am simply ignorant, and others might suffer from this problem too. A simple internet search indicated that A. oryzae has more common uses. Mostly these involve the production of sake, miso, and soy sauce. How do I get from the production of sake to insect pathogen? A one-paragraph overview of A. oryzae would help.

3) It has been suggested that A. oryzae is a mutant of A. flavus (https://doi.org/10.1080/13693780600835716). At the end, are you creating something new or are you reactivating aflatoxin producing genes that have been silenced in the domestication process?

Especially in section headings) italicize A. oryzae and L. migratoria no matter where these are used.

Line 108) In measuring colonies, how did you prevent contamination?

Line 114) Excel does not do Probit analysis (at least mine does not). Is this a custom program written in Excel?

Line 122) correct English. Consider breaking into two or more sentences.

Figure 3) TQ541 is in black. However, in Figure 4 it is red. It would be nice to have a consistent color.

Figure 4A) How does Figure 4A differ from Table 2? If they are the same, then one can be removed.

Line 174) Given that 24C is suboptimal, why do we care about this?

Table 3) This is useful as it is an independent demonstration that 33C is at or near the optimal growing conditions.

Line 189) … generated by …

Line 190) separate sentences.

Line 218) You are claiming that A. oryzae is adapted to a broad temperature range. What is a more typical temperature range for other entomopathogenic fungi (or other fungi more generally)?

The supplementary figure indicates that you have missed something. Genetically, four of the five mutants are identical. The only different one is TQ302. So given the genetic identity with the mutants TW201, TQ238, and TQ555, what is special about TW541?

Comments on the Quality of English Language

Author Response

Reviewer 3

The development of new insect management methodologies is an important part of the continued production of high-quality agricultural products. Developing a new biocontrol organism through enhanced mutation is certainly worth exploring.

A: Thank you very much.

There are a few things that should be covered in the introduction to make this paper easier and more informative to readers.

Q1) Is space radiation less harmful than other radiation sources? Is there something about Space that makes this approach worth the extra cost versus exposing them to a radiation source closer to home?

A: Thanks for your valuable suggestions. Space radiation is as harmful as other radiation sources, but Space mutagenesis can generate high numbers of mutations and yield variable and stable mutants. We add the sentences “The vast, cold, and radiation-filled conditions of outer space pose major challenges for all organisms [23]. Microorganisms rapidly adapt to environmental changes by altering the expression of their genes [24].” in line 60-62 in clear version.

Horneck, G.; Klaus, D.M.; Mancinelli, R.L. Space Microbiology. Microbiol. Mol. Biol. Rev. 2010, 74, 121–156.

Acres, J.M.; Youngapelian, M.J.; Nadeau, J. The influence of spaceflight and simulated microgravity on bacterial motility and chemotaxis. NPJ Microgravity 2021, 7, 7.

Q2) This is the first I have seen Aspergillus oryzae introduced as an insect biocontrol agent. Sometimes I am simply ignorant, and others might suffer from this problem too. A simple internet search indicated that A. oryzae has more common uses. Mostly these involve the production of sake, miso, and soy sauce. How do I get from the production of sake to insect pathogen? A one-paragraph overview of A. oryzae would help.

A: Thanks for your valuable suggestions. We add sentences “Generally, Aspergillus oryzae is one major component of the leaven in traditional fermented food such as soy sauce, soybean paste and rice wine in China, Japan and other Asian countries for centuries [16]. The pathogenicity of Aspergillus to locust is rarely reported. A. flavus had high virulence to desert locust Schistocerca gregaria [17].” in line 43-46 in clear version.

Zhong, Y.Y.; Lu, X.; Xing, L.; Ho, S.W.A.; Kwan, H.S. Genomic and transcriptomic comparison of Aspergillus oryzae strains: a case study in soy sauce koji fermentation. Journal of Industrial Microbiology & Biotechnology. 2018, 45(9), 839-853.

Venkatesh, M.V.; Joshi, K.R.; Harjai, S.C.; Ramdeo, I.N. Aspergillosis in desert locust (Schistocerka gregaria Forsk). Mycopathologia. 1975, 57(3), 135-8.

3) It has been suggested that A. oryzae is a mutant of A. flavus (https://doi.org/10.1080/13693780600835716). At the end, are you creating something new or are you reactivating aflatoxin producing genes that have been silenced in the domestication process?

A: Thank you very much. I will answer this question from two aspects as below:

  • Really, it has been suggested that oryzae is a domesticated ecotype of A. flavus and does not produce aflatoxin. We had analyzed the production of aflatoxin for wildtype A. oryzae XJ-1 and the results indicated that it could not produce aflatoxin B1 (Zhang et al., 2015).

Zhang, P.; You, Y.; Song, Y.; Wang, Y.; Zhang, L. First record of Aspergillus oryzae (Eurotiales: Trichocomaceae) as an entomopathogenic fungus of the locust, Locusta migratoria (Orthoptera: Acrididae). Biocontrol Sci. Techn. 2015, 25, 1285-1298.

  • It is reported that AflR is a regulator in the biosynthesis of aflatoxin. If aflR gene is not expressed, it won’t produce aflatoxin in an Aspergillus strain (Kobayashi et al., 2007).

Kobayashi T, Abe K, Asai K, Gomi K, Juvvadi PR, Kato M, Kitamoto K, Takeuchi M, Machida M. Genomics of Aspergillus oryzae. Biosci Biotechnol Biochem. 2007, 71(3): 646-70.

We have sequenced the genome of A. oryzae XJ-1, and there is no aflR gene in the genome of A. oryzae XJ-1. We also sequenced three transcriptomes of A. oryzae XJ-1 by different treatments, and there is no expression of aflR gene in the transcriptomes of A. oryzae XJ-1. We also sequenced the genomes of these five mutants (TQ201, TQ238, TQ302, TQ549, and TQ555), and there are no aflR gene in the genomes of these mutants. We think A. oryzae XJ-1 don’t produce aflatoxin, the main reason is no aflR gene in the genome. Since there are also no aflR gene in the genomes of these five mutants, we think we don’t reactivate aflatoxin producing genes. Because we are writing another paper about genomes and transcriptomes of A. oryzae XJ-1, we don’t show these results in this paper.

Especially in section headings) italicize A. oryzae and L. migratoria no matter where these are used.

Answer: Thanks for your valuable suggestions. We have corrected this.

Line 108) In measuring colonies, how did you prevent contamination?

Answer: Line 120. Thank you very much. Actually, we measured colony diameters every day by a ruler from the back of culture dishes until the 8th day after inoculation. We did not open the PDA plate during the measurement. This method won’t result in contamination.

So we changed the sentence “The growth rates of TQ549 and A. oryzae XJ-1 were evaluated by measuring colony diameters every day by a vernier caliper until the 8th day after inoculation” into the sentence “The growth rates of TQ549 and A. oryzae XJ-1 were evaluated by measuring colony diameters every day with a ruler from the back of culture dishes until the 8th day after inoculation.” in line 117-119 in clear version.

Line 114) Excel does not do Probit analysis (at least mine does not). Is this a custom program written in Excel?

Answer: Line 128. Thank you very much. Excel software has this function and we have used this method to get LT50 and LC50 of wildtype A. oryzae XJ-1 against adult locusts and published a paper this year (You et al., 2023).

You, Y; An, Z; Zhang, X; Liu, H.; Yang, W.; Yang, M; Wang, T; Xie, X; Zhang, L. Virulence of the fungal pathogen, Aspergillus oryzae XJ-1 to adult locusts (Orthoptera: Acrididae) in both laboratory and field trials. Pest Manag. Sci. 2023, 79, 3767-3772.

Line 122) correct English. Consider breaking into two or more sentences.

Answer: Line 135. Thanks for your valuable suggestions. We have changed this sentence into “they eventually covered the bodies of dead locusts, especially TQ549 mainly exhibited white color (Figure 1)”in line 132-134 in clear version.

Figure 3) TQ549 is in black. However, in Figure 4 it is red. It would be nice to have a consistent color.

Answer: Thanks for your valuable suggestions. We have adjusted TQ549 from black to red in Figure 3, which is the same color in Figure 4 as you suggested.

Figure 4A) How does Figure 4A differ from Table 2? If they are the same, then one can be removed.

Answer: Thanks for your valuable suggestions. They are the same, and we removed Table 2 according to your suggestion.

Line 174) Given that 24 C is suboptimal, why do we care about this?

Answer: Line 198. The main purpose is to explore the scope of application of Aspergillus oryzae XJ-1. In the Northwest of China, such as the Inner Mongolia Autonomous Region and the Xinjiang Uygur Autonomous Region, main locust breeding regions in China, when the locust plague breaks out in the summer, the temperature was about 24℃ due to the high altitude.

Table 3) This is useful as it is an independent demonstration that 33C is at or near the optimal growing conditions.

Answer: Thank you very much.

Line 189) … generated by …

Answer: Line 214. Thanks for your valuable suggestions. We have corrected this as your suggestion. We changed the sentence into “Five mutants were screened from 564 strains of A. oryzae XJ-1 generated by space mutagenesis,” in line 198-199 in clear version.

Line 190) separate sentences.

Answer: Line 214. Thanks for your valuable suggestions. We have separated this sentence into two sentences “Five mutants were screened from 564 strains of A. oryzae XJ-1 generated by space mutagenesis. The colony morphological characteristics of these five mutants on PDA plates differed from that of A. oryzae XJ-1” in line 198-200 in clear version.

Line 218) You are claiming that A. oryzae is adapted to a broad temperature range. What is a more typical temperature range for other entomopathogenic fungi (or other fungi more generally)?

Answer: Line 222. Thank you very much. For another important locust fungal pathogen Metarhizium anisopliae var. acridum, its optimal temperature is 28 °C. It could effectively kill locust at the temperature range from 20 to 30 °C. Once the temperature is over 30 °C, its capacity will significantly decrease (Arthurs and Thomas, 2001). But for A. oryzae XJ-1, it could effectively control locusts even at high temperature 38 °C by our field experiment (You et al., 2023). In this paper, we find that A. oryzae XJ-1 and TQ549 cloud rapidly grow at temperature 24 and 38 °C. So we claim that A. oryzae XJ-1 and TQ549 are adapted to a broad temperature range.

Arthurs S, Thomas MB. Effects of temperature and relative humidity on sporulation of Metarhizium anisopliae var. acridum in mycosed cadavers of Schistocerca gregaria [J]. Journal of Invertebrate Pathology, 2001, 78(2): 59-65.

You, Y; An, Z; Zhang, X; Liu, H.; Yang, W.; Yang, M; Wang, T; Xie, X; Zhang, L. Virulence of the fungal pathogen, Aspergillus oryzae XJ-1 to adult locusts (Orthoptera: Acrididae) in both laboratory and field trials. Pest Manag. Sci. 2023, 79, 3767-3772.

The supplementary figure indicates that you have missed something. Genetically, four of the five mutants are identical. The only different one is TQ302. So given the genetic identity with the mutants TQ201, TQ238, and TQ555, what is special about TQ549?

Answer: Thank you very much. Generally, ITS sequence might be used to identify strain to genus and species. We screen the mutants by the morphology on PDA plates. The significant character of TQ549 is that its colony is white.
